# Initiator-Directed Transcription: Fission Yeast Nmtl Initiator Directs Preinitiation Complex Formation and Transcriptional Initiation

**DOI:** 10.3390/genes13020256

**Published:** 2022-01-28

**Authors:** Diego A. Rojas, Fabiola Urbina, Lucía Valenzuela-Pérez, Lorenzo Leiva, Vicente J. Miralles, Edio Maldonado

**Affiliations:** 1Instituto de Ciencias Biomédicas, Facultad de Ciencias de la Salud, Universidad Autónoma de Chile, Santiago 8910132, Chile; 2Programa de Biología Celular y Molecular, ICBM, Facultad de Medicina, Universidad de Chile, Santiago 8380492, Chile; fabiola.urbina1516@gmail.com (F.U.); lucia.valenzuela@uchile.cl (L.V.-P.); leiva.lorenzo.e@gmail.com (L.L.); 3Departamento Bioquímica y Biología Molecular, Facultad de Farmacia, Universidad de Valencia, 46010 Valencia, Spain; vicente.j.miralles@uv.es

**Keywords:** initiator, transcription, general transcription factors (GTFs), RNA polymerase II, *Schizosaccharomyces pombe*

## Abstract

The initiator element is a core promoter element encompassing the transcription start site, which is found in yeast, *Drosophila*, and human promoters. This element is observed in TATA-less promoters. Several studies have defined transcription factor requirements and additional cofactors that are needed for transcription initiation of initiator-containing promoters. However, those studies have been performed with additional core promoters in addition to the initiator. In this work, we have defined the pathway of preinitiation complex formation on the fission yeast nmt1 gene promoter, which contains a functional initiator with striking similarity to the initiator of the human dihydrofolate reductase (hDHFR) gene and to the factor requirement for transcription initiation of the nmt1 gene promoter. The results show that the nmt1 gene promoter possesses an initiator encompassing the transcription start site, and several conserved base positions are required for initiator function. A preinitiation complex formation on the nmt1 initiator can be started by TBP/TFIIA or TBP/TFIIB, but not TBP alone, and afterwards follows the same pathway as preinitiation complex formation on TATA-containing promoters. Transcription initiation is dependent on the general transcription factors TBP, TFIIB, TFIIE, TFIIF, TFIIH, RNA polymerase II, Mediator, and a cofactor identified as transcription cofactor for initiator function (TCIF), which is a high-molecular-weight protein complex of around 500 kDa. However, the TAF subunits of TFIID were not required for the nmt1 initiator transcription, as far as we tested. We also demonstrate that other initiators of the nmt1/hDHFR family can be transcribed in fission yeast whole-cell extracts.

## 1. Introduction

RNA polymerase II (RNAPII) is a multimeric enzyme that transcribes protein-coding genes, microRNAs, long non-coding RNAs, and small nuclear RNAs. Despite being a multimeric enzyme, RNAPII is unable to recognize gene promoters and start transcription initiation in the absence of additional factors. These set of additional factors are called general transcription factors (GTFs), and include TFIIA, TFIIB, TFIID (TBP + TAFs), TFIIE, TFIIF, and TFIIH, defined as the minimal set of GTFs required to in vitro transcribe a gene containing the TATA box promoter element. The GTFs are required for transcription of most, if not all, the genes transcribed by RNAPII, and they are evolutionarily well conserved from yeast to humans. On the other hand, RNAPII needs promoter elements for accurate start site selection and transcription initiation [1,2]. RNAPII promoters are composed of multiple elements, known as the core promoter elements (CPE), which are necessary for start site selection and preinitiation complex (PIC) formation [1,2]. The most important CPEs in metazoan cells include: the TATA box, the B recognition element (BRE), the downstream promoter element (DPE), the initiator (Inr), the motif ten element (MTE), the downstream core element (DCE), the TCT motif, the X core promoter elements 1 and 2 (XCPE1/XCPE2), and the homology D box (HomolD box) [3,4,5]. The CPEs that can direct transcription initiation in fission yeast are: the TATA box, the Inr element, the TCT motif, the XCPE1/XCPE2 elements, and the HomolD box. The BRE, DPE, MTE, and DCE cannot direct transcription initiation alone; however, most of them are found associated with the TATA box or the Inr elements, and serve as regulatory elements.

The Inr is a CPE that was discovered in an analysis of the gene promoter of lymphocyte-specific terminal transferase (TdT) [6]. This CPE is analogous in function to the TATA box, since it can direct transcription initiation; however, it is usually located between −3 and +5 relative to the transcription start site (TSS), where +1 is the TSS. The mechanisms that govern PIC formation in TATA-less promoters are still unknown, particularly in Inr-containing promoters, although a great amount of information has been obtained from mutational analysis on the Inr [7,8]. The protein factor(s) that specifically recognize the Inr are not known yet, although is believed that TAFs of the transcription factor TFIID are able to recognize the Inr in order to start the assembly of a PIC competent to start transcription [9,10,11]. An in vitro transcription assay with pure RNAPII and recombinant GTFs has not yet been set up, neither has stable binding using electrophoretic mobility shift assays (EMSA) been obtained using the Inr element, purified recombinant GTFs, and RNAPII. Moreover, the consensus sequence of the human Inr is still a matter of controversy [12]. By using mutational analysis, the consensus sequence for the human Inr was defined as YYA + 1NWYY (−2 to +5), where Y = C/T, W = A/T, N = A/C/G/T, and A + 1 is the initiation site [7,8]. However, later use of genome-wide mapping via cap analysis gene expression (5′CAGE) led to a very short Inr consensus sequence, YR (−1 to +1), where R = A/G and A + 1 is the initiation site [13]. Recently, a more accurate analysis has led to the Inr consensus sequence of BBCA + 1BW (−3 to +3), where B = C/G/T, W = A/T, and A + 1 is the initiation site [14]. This consensus sequence resembles the Inr consensus sequence defined by mutational analysis.

In the fission yeast *Schizosaccharomyces pombe*, DeepCAGE analysis has also been performed to find CPEs, which are present in gene promoters [15]. It was found that canonical TATA boxes were centred around −28 to −30, and an Inr consensus sequence around the TSS, which had thesequence PyPyPu + 1N(A/C)(A/C), where Py = A/G, Pu = C/T, N = A/C/G/T, and the initiation site was Pu + 1 [15]. This sequence is much closer to the human Inr consensus sequence than to the *S. cerevisiae* Inr.

The nmt1 gene in *S. pombe* is highly transcribed and completely repressed by thiamine [16]. It possesses a canonical TATA box located at −25 to −30 base pairs upstream of the TSS; however, deletion of the TATA box does not abolish transcription, it only produces a small diminution of transcription [17]. The TSS is not altered when the TATA box is mutated or deleted [17,18], and a comparison of the region surrounding the nmt1 gene TSS revealed that this region might contain a functional Inr (ATCA + 1ATTG). We thought that the transcription initiation region of the nmtl gene might contain a functional Inr, since it matches the mutated human Inr consensus sequence YYA + 1NWYN. By using a series of point mutations in the nmt1-transcription initiation region, we demonstrated that this region contains a functional Inr, which is required to direct PIC formation and drive transcription initiation. We also demonstrated that the first step of PIC formation is the binding of TBP/TFIIA or TBP/TFIIB to the nmt1 Inr, followed by the recruitment of the TFIIF/RNAPII complex. Following these initial steps is the recruitment of the rest of the GTFs; however, the PIC is not competent for transcription initiation, and additional protein factors present in crude whole-cell extracts are needed to enable transcription. We were able to extend these observations to other nmt1 Inr-related gene promoters, which contained a similar Inr at the transcription initiation region.

## 2. Materials and Methods

### 2.1. Purification of GTFs, PC4, RNAPII and Mediator

Recombinant GTFs (TFIIA, TBP, TFIIB, TFIIE, TFIIF, and PC4) were expressed and purified as described in earlier studies [19,20]. TFIIH was purified from whole-cell extracts (WCE) from a TAP-tagged p62 subunit (Tfb1) strain by incubating cell extracts with IgG-Sepharose beads, with subsequent washing and eluting with TEV protease. RNAPII core enzyme was purified from a wild-type strain using conventional chromatography as described previously [19,20]. Mediator was purified from WCE from a fission yeast TAP-tagged Med7 strain by incubating cell extracts with IgG-Sepharose resin, then washing and eluting the bound complex by treatment with TEV protease [19,20].

### 2.2. Mediator and TAF Depletion

Mediator was depleted from a WCE of a TAP-tagged Med7 strain, by adjusting the WCE at 0.5 M potassium acetate and incubating two volumes of settled IgG-Sepharose with one volume of WCE. The mix was incubated by rocking for 30 min at room temperature, and then for 2 h at 4 °C. The mix was centrifugated and the supernatant used as depleted WCE (WCED). TAFs were depleted using a similar approach, except they were incubated with a column that contained 1 mg/mL of rabbit anti-TAF1 crosslinked to CNBr-activated Sepharose.

### 2.3. Electrophoretic Mobility Shift Assays

EMSA assays were performed as described previously [19]. Each binding reaction contained the following binding mix: 20 mM HEPES (pH 7.9), 50 mM KCl, 5 mM MgCl2, 0.1 mM EDTA, 5% glycerol, 0.5% PEG 8000 (Sigma-Aldrich, St. Louis, MO, USA), 2 mM DTT, 0.1 mM PMSF, 50 ng BSA, and 50 ng of poly(dI-dC) or poly(dG-dC). Recombinant and purified GTF and RNAPII were used in the assays. Proteins were incubated with binding mix for 5 min at 25 °C. Next, 5–10 ng of Inr-containing probe labeled with ^32^γATP was added to the assays, and the reaction mixes were incubated for 15 min at 30 °C. Inr-containing probe with single mutations was evaluated in the same manner as the wild-type probe. The DNA-protein complexes were evaluated in 5% acrylamide gels containing 10% glycerol, and run at 100 V at 4 °C for 2 h in 50 mM Tris-borate (pH 8.3) buffer. EMSA gels were prerun at 4 °C for 1 h. Detection of the complexes was performed using autoradiography analysis.

### 2.4. In Vitro Transcription Assays

In vitro transcription was performed as described previously [19,20] using 100 ng of nmt1 gene promoter containing a TATA box (wt TATA Inr) or Inr (wt Inr) template DNA. In some assays, several Inr-containing templates with single mutations were evaluated. These constructs were made using gene synthesis at Genscript, Inc (Piscataway, NJ, USA). Each wild-type or mutated synthetic promoter fused to a 360 bp G-less cassette was synthetized, with an EcoRI restriction site at the 5′ end and a BamHI site at the 3′ end, then digested and ligated into the EcoRI and BamHI sites of pUC58 vector. Three clones for each construct were analyzed and sequenced at Genscript. The plasmid DNA used for transcription reactions was purified using the E.Z.N.A. Plasmid Midi Kit according to manufacturer instructions (Omega Bio-tek, Norcross, GA, USA). The promoters fused to the G-minus cassette upon digestion of the transcripts by RNase T1, producing a transcript of 370 nucleotides. Reactions were performed with 5 μL of WCE (10 mg/mL) or WCED (Mediator-depleted extract). In some assays, α-amanitin was added in concentrations of 0.5, 1, 2, 4, and 8 μg/mL. Transcript detection was performed using autoradiography analysis on an X-ray film.

### 2.5. Purification of TICF

TICF was purified from wild-type fission yeast whole-cell extracts. Extracts were fractionated on a phosphocellulose (Whatman P11) column equilibrated in buffer A (25 mM HEPES pH 7.9, 50 mM KCl, 0.1 mM EDTA, 2.5 mM DTT, 10% glycerol, 0.1 mM PMSF). After loading, the column was washed with buffer A and sequentially eluted with 0.3, 0.5, and 1.0 M of KCl in buffer A. The fraction containing the transcription complementing activity (0.5 M) was dialysed against buffer A and loaded onto a heparin-agarose column and eluted as described for the P11 column. The fraction that contained the transcription complementing activity (0.5 M) was dialysed against 25 mM potassium phosphate buffer pH 7.9, 2.5 mM DTT, 10% glycerol, and 0.1 mM PMSF and loaded onto a hydroxyapatite column equilibrated in dialysis buffer. The column was washed and eluted with a potassium phosphate gradient (50–400 mM), and the transcription complementing activity eluted as a single peak at 100 mM phosphate, and separated from RNAPII and transcription factors. Subsequently, the fractions containing activity were pooled and dialysed against buffer A and loaded onto a Q-Sepharose column, washed and eluted with a gradient of KCl (50–400 mM) and the transcription complementing activity eluted as a single peak at 150 mM KCl. Active fractions were pooled and concentrated against buffer A containing 55% glycerol and loaded onto a AcA22 gel filtration column, then separated in buffer A containing 1 M KCl. The activity eluted as a single peak at approximately 500 kDa MW, and was devoid of GTFs, TAFs, Mediator, and CK2, as determined by Western blot analysis. These active fractions were stored at −80 °C and used in transcription assays.

## 3. Results

### 3.1. The Transcription Start Point of the Nmtl Promoter Has Homology to the Human Dihydrofolate Reductase Inr

The nmtl1 gene promoter of *S. pombe* has been described elsewhere [16,17,18]. This gene promoter contains a classical TATA box, located at −25 from the TSS; however, it has been observed that mutation or deletion of this CPE does not abolish in vivo transcription and the TSS remains unchanged. Deletion of the TATA box causes a decrease in transcription, indicating that sequences around the TSS should be able to direct transcription initiation. Careful examination of the nmtl TSS region revealed that there is a region around the TSS with striking homology to the human dihydrofolate reductase (hDHFR) core promoter, which is well characterized and a model upon which to study transcription from TATA-less promoters [19,20]. Moreover, a search for *S. pombe* in the new Eukaryotic Promoter Database (EPD) indicates that several TATA-less promoters contain a highly similar sequence, which we identify as the nmt1/hDHFR Inr family (Figure 1A and for a complete list see Appendix A). Several positions are conserved, such as −1 (C), +1 (A), +2 (A), and +4 (T), which are most likely important for Inr function (see below). A search for promoters in *S. pombe* from the EPD (selecting from −30 to +20), with the consensus sequence TTCAACTT, produces a strong peak around the TSS, indicating that the sequence is highly represented in the EPD (Figure 1B).

### 3.2. A TATA Box Mutated Nmtl Promoter Is Transcriptionally Active in S. pombe Whole-Cell Extracts

To investigate whether the nmt1 promoter contains a functional Inr around the TSS, a fragment containing the wild-type promoter from −40 to +6 was fused to the G-less cassette (G +5 was changed to C) and in vitro transcribed in fission yeast whole-cell extracts (WCE). The results obtained were compared with a similar template, in which a TATA box mutated nmt1 promoter (from −20 to +6) was used. The results of these experiments show that there was a concentration dependence of α-Amanitin on transcription inhibition of the evaluated promoters. In the case of wild-type TATA-containing promoter, this promoter was transcribed in WCE and inhibited by 4 μg/mL α-Amanitin, indicating that transcription is carried out by RNAPII (Figure 2). Likewise, the TATA-lacking nmt1 promoter was transcribed in the WCE and the transcription was carried out by RNAPII (Figure 2), which is sensitive to 2 μg/mL of α-Amanitin, indicating that sequences around the TSS contain a functional Inr that can direct transcription initiation. It should be noted that the TATA-lacking nmt1 promoter was more sensitive to α-Amanitin than the TATA-containing promoter, probably due to the fact that the TATA-lacking promoter is weaker than the wild-type TATA-containing promoter. Nevertheless, both promoters were sensitive to α-Amanitin, indicating that they were transcribed by RNAPII.

### 3.3. Mutations around the TSS of nmt1 Promoter Can Impair Transcriptional Initiation

To define the necessary region for nmt1 promoter transcription initiation, we mutated the region surrounding +1, including the +1 transcription start site. Mutations were performed according to the conserved positions in the nmt1 Inr family. The constructs carrying the mutant promoters were transcribed in vitro in *S. pombe* WCE. The results obtained from these experiments are shown in Figure 3. Single point mutations in the conserved bases in the TSS almost completely abolished transcription, indicating that this region is responsible for directing transcription initiation from the nmtl TATA-lacking promoter (Figure 3, positions +1 and +2). Point mutations in conserved positions outside of +1 also had a strong effect (Figure 3, positions −1 and +4), suggesting that they are important for nmt1 Inr function. Point mutations in positions −2 and +3 did not seem to completely abolish transcription initiation from the nmt1 Inr, indicating that some variation could occur in these positions, or that pyrimidines were changed to other pyrimidines, a difference from positions +1 and +2, or purines were changed to pyrimidines, in an effort to change and inactivate the Inr consensus sequence.

Taken together, these results indicate that the nmtl promoter possesses a bona fide Inr around the TSS, which can direct transcription initiation in the absence of a TATA-box, as has been observed under in vivo conditions [17,18].

### 3.4. Nmtl Inr-Directed Transcription Initiation Is Mediator-Dependent but Is TAF-Independent

We performed these experiments using TAFs and Mediator-depleted *S. pombe* WCE (WCED). TAFs were depleted using anti-TAF1 antibodies crosslinked to Sepharose beads, and Mediator was depleted from a TAP-tagged Med7 strain using IgG Sepharose beads. TAF-depletion did not affect the transcription from the nmt1 TATA-lacking Inr-containing promoter (Figure 4A). However, depletion of Mediator completely abolished transcription from the same promoter (Figure 4A). Transcription initiation can be restored by the replacement of the protein fraction eluted from the IgG Sepharose column, but not from the protein fraction eluted from the anti-TAF1 column. Neither a combination of GTFs plus PC4, or GTFs plus PC4 plus RNAPII, could restore transcription, indicating that these combinations of factors and RNAPII cannot replace Med function (Figure 4A). A crude fraction (0.5 M P11) from WCE, which was fractioned via phosphocellulose chromatography and contained GTFs, PC4, Med, and RNAPII, could restore transcription of the nmt1 promoter in the WCED. Western blot analysis indicated that most of the Med complex was depleted from the WCE by the IgG Sepharose column (panel Med17, Figure 4B), and most of the TAF1 and TAF5 polypeptides were depleted using anti-TAF1 chromatography (panel TAF1 and TAF5, Figure 4B). Neither the IgG Sepharose column or the anti-TAF1 column could deplete TAF or Mediator complexes, respectively, since TAF5 was present in the WCED via IgG Sepharose beads, and Med17 was present in the WCED via anti-TAF1 Sepharose beads (Figure 4B). However, we cannot rule out that other TAF-containing complexes were still present in the anti-TAF1 depleted WCE, since a high amount of TBP was still present in the depleted extract (panel TBP, Figure 4B). On the other hand, most of the Med complex can be removed using the depletion procedure, as the Med17 polypeptide can be completely removed via chromatography on IgG Sepharose beads. RNAPII and other tested GTFs were not significantly depleted by either IgG Sepharose or anti-TAF1 Sepharose beads (Figure 4B). We can conclude that a TAF1-containing complex is not necessary for transcription directed by the nmt1 TATA-lacking Inr-containing promoter; however, the Med complex is required to support in vitro transcription from this promoter.

### 3.5. A Dimeric Complex between TFIIA-TFIIB or TBP-TFIIB Recognizes the Nmtl Inr

To investigate whether the basal transcription machinery itself is able to recognize the nmtl Inr, we used a gel retardation assay set up with 5 ng of fission yeast TBP and increasing amounts of recombinant fission yeast TFIIA or TFIIB (Figure 5A), since those factors bind in the first step of PIC formation on TATA-containing promoters. The results shown in Figure 5A indicate that TBP alone cannot securely bind to the nmt1 Inr; however, both TFIIA or TFIIB can induce the stable binding of TBP to the Inr element. A combination of TFIIA and TFIIB did not form a stable complex on the nmt1 Inr (Figure 5A), indicating that TBP is the key transcription factor able to bind the nmt1 Inr and start PIC complex formation. The binding of these factors to the nmt1 Inr is specific, since a mutated Inr (at +1 or +2), which cannot drive transcription initiation, is unable to form a PIC (Figure 5B). Furthermore, point mutations in −1 or +4 can abolish PIC formation; however, a point mutation in +3 (Figure 5B), which reduces transcription initiation, does not interfere with PIC complex formation, indicating that this mutation might be able to interfere with a downstream step of the Inr-TBP-IIB complex formation. Once TBP-TFIIB are bound to the promoters a conventional PIC formation follows, succeeded by the binding of RNAPII-TFIIF and TFIIE (and TFIIH) recruitment into the PIC (Figure 5C).

### 3.6. In Vitro Transcription Reconstitution from the Nmtl Inr Promoter

We sought to investigate the necessary factors for Inr-dependent transcription using a reconstituted transcription system with purified RNAPII, TFIIH, Med complex, and recombinant TFIIA, TBP, TFIIB, TFIIE, TFIIF, and PC4 (Figure 6A and Appendix A). However, although nmt1 Inr promoter can form a PIC, this is not competent for transcription initiation (Figure 5 and Figure 6A), whereas the TATA box of the Ad-MLP promoter is able to form a PIC with the RNAPII and GTFs and is fully competent to initiate transcription.

Complementing activities that can reconstitute Inr transcription were found in WCEs in the 0.5 M KCl chromatographic fraction of a phosphocellulose column (Figure 6A,B). We further fractionated the activity on heparin-agarose, Q-Sepharose, and hydroxyapatite, and assayed their activity using transcription assays (Figure 6C). Transcriptionally active fractions from the hydroxyapatite column were pooled and fractionated onto a A_c_A_22_ gel filtration column in high-salt buffer. The activity elutes in a single peak of 500 kDa MW, indicating that it is a complex composed of several subunits (Figure 6C). Western blot analysis indicated that the high MW fraction is devoid of RNAPII, Med subunits, TAF1-like complex, TBP, CK2, and GTFs; however, it still contained several proteins, and we have not yet been able to identify the polypeptides (data not shown). We speculate that this high MW fraction is composed of transcriptional coactivators, which can interact with GTFs, RNAPII, or Med subunits. We refer to this fraction as TCIF (transcriptional cofactor of Inr function), and believe it is required to reconstitute the transcription activity from the nmt1 Inr, together with the GTFs, RNAPII, Mediator, and PC4 (Figure 6D). PC4 was required in this assay, since in its absence no transcript was obtained (Figure 6D, lane -PC4 + TCIF).

### 3.7. TATA-Lacking Promoters with a Promoter of the nmt1 Inr/hDHFR Family Can Direct Transcription in Fission Yeast WCE

We sought to investigate whether TATA-lacking promoters of the nmt1/hDHFR Inr family can also direct transcription in fission yeast WCE. We cloned three additional promoters (FP000624, FP000463, and FP000396) from −20 to +6, and they were assayed and compared with nmt1 Inr and a mutated nmt1 Inr (+4) in transcription in a WCE. As expected, those promoters were able to direct transcription initiation in WCE (Figure 7), although the FP000463 promoter was not as strong as the nmt1 Inr, FP000624, and FP000396 promoters. These results indicate that Inrs of the nmt1/hDHFR family are strong CPEs that can function in the absence of other upstream or downstream CPEs.

## 4. Discussion

In this work, we demonstrate that the nmtl gene possesses an Inr encompassing the transcription start site. This Inr is capable of directing PIC formation and transcription initiation. Nmtl Inr-directed transcription requires the GTFs, RNAPII, Med, PC4, and a high-molecular-weight protein fraction that we termed TICF. We have not yet identified any polypeptide from TICF, although we speculate that it contains protein kinases and transcriptional coactivators. However, we have determined, using Western blot analysis, that the fraction is free of GTFs, RNAPII, Med, TAF, and protein kinase CK2. It is well known that in vitro basal transcription from TATA-containing promoters requires RNAPII and GTFs, including TFIIA, TFIIB, TBP, TFIIE, TFIIF, and TFIIH [21,22], although TFIIH is not required on supercoiled templates [23,24]. Most of the biochemical studies dissecting the transcriptional mechanism have been conducted using the Ad-MLP promoter; therefore, our knowledge of transcriptional initiation is limited, in part because the Ad-MLP promoter contains a strong TATA box. However, only a minority (<25%) of RNAPII promoters from the human genome have a TATA box, and approximately 50% contain an Inr consensus sequence around the TSS [13,14,25]. Recently, it has been found that 40% of the human focused RNAPII core promoters contain an Inr motif, which closely matches the BBCABW consensus sequence [14]. In fission yeast, a recent genome-wide study using the learning framework DeepCAGE has revealed that only 8% of the RNAPII core promoters contain a TATA box, whereas almost 90% might contain an Inr-like motif around the TSS [15]. These observations indicate the need to study the Inr-containing promoters in both human and fission yeast. However, the GTFs plus RNAPII are unable to direct transcription from TATA-less promoters.

Despite the current knowledge of the factors required for TATA-box-mediated transcription, the requirement of protein factors for transcription initiation from TATA-lacking promoters remains poorly defined. Several studies that aimed to identify the minimal components used a TATA box in the context of an Inr to identify factors, which, together with GTFs and RNAPII, could allow Inr activity. Kaufmann et al. identified from HeLa nuclear extracts a protein fraction known as CIF (cofactor of Inr function), which can stimulate Inr activity in the presence of GTFs [26,27]. This protein fraction has multiple components, one of which is TAF2 [26]. Indeed, another related study found that a dimer of TAF1/TAF2 can recognize the Inr, and mutations within the Inr that impair the binding of TAF1/TAF2 also impair transcription initiation [28]. In *Drosophila*, a trimeric TBP/TAF1/TAF2 can support Inr activity when added in place of TFIID [29]. However, in humans, TFIID that can bind to the Inr in vitro and support Inr activity lacks a TAF2 homologue [11,30]. In a different study, Reinberg and colleagues [31] set out to perform an assay using a promoter containing an upstream Sp1 binding site, the β-globin Inr, and a DPE element, and found that TAFs, Mediator, PC4, and protein kinase CK2 are required, in addition to GTFs and RNAPII, to establish transcription from that promoter. To reiterate, the multiple CPEs in these promoters could impose additional factor requirement for transcription. More recently [32], HMGA1 has been identified as a link between Med and TAFs in an assay dependent on TATA and Inr core promoter elements. TAFs are required to counterattack the transcriptional negative effects of Topoisomerase I and NC2 (DR1/DRAP1) [32]. In our fission yeast system, an Inr is the unique CPE directing transcription initiation, and its transcription requires GTFs, RNAPII, PC4, Mediator, and TICF; however, transcription is completely TAF-independent. We speculate that fission yeast Inrs of the nmt1/hDHFR family are stronger CPEs than Inrs from humans or *Drosophila*. Indeed, a report from Conaway and colleagues has shown that the hDHFR promoter (a TATA-less promoter) can direct PIC formation and transcription initiation in a reconstituted system, and although transcription is highly dependent on TFIID, TBP can still direct low transcription levels [33]. It is possible that, since TFIID and TFIIH were purified via conventional chromatography from rat liver extracts, they could have been contaminated with small amounts of TICF-like activities. Nevertheless, we believe that the promoter (hDHFR) and the developed transcription system is a good model for studying the molecular mechanisms governing transcription in TATA-less promoters, since hDHFR has been extensively studied as a model for TATA-lacking promoters [33,34].

The Inr seems to be a widespread CPE, found in fission yeast, humans, insects, and many other metazoan organisms [35,36]. Indeed, it has been proposed that the Inr is an ancient CPE, since it is also found in ancient eukaryotes, such as *Trichomonas vaginalis* [37]. RNAPII core promoters in *T. vaginalis* lack a TATA-box; however, they contain an Inr motif as the sole CPE. This element is bound by IBP39, which is an initiator-binding protein in *Trichomonas* [38]. It is believed that IBP39 can nucleate a PIC in *Trichomonas*, since it can interact with the CTD of the largest subunit of *Trichomonas* RNAPII [39]. Indeed, two reported Inrs from *T. vaginalis* are strikingly related to the nmt1/hDHFR Inr family (pol II: TATCAAAATAAT and TVCA1: TCTCAAATTTT) [35,36]. IBP39 does not display homology to any other transcription factor in the database and is exclusive of the *Trichomonas* genus. Furthermore, two TBPs have been identified, one of which binds to *T. vaginalis* gene promoter regions, and the other (TvTBP1) interacts with IBP39, suggesting that they might be part of a PIC at RNAPII core promoters [40].

From our present study of Inr-directed transcription in fission yeast, it is clear that TAFs do not play a role in transcription initiation from Inr-containing TATA-lacking promoters in *S. pombe*, although in metazoan TATA-less promoters, such as in humans and insects, TAFs seem to be essential for transcription initiation [26,27,28,29,30]. This difference could be a consequence of the Inr promoter strength, since Inrs from fission yeast could be stronger than metazoan Inrs. On the other hand, Med is required for transcription initiation in crude extracts [41,42,43], and for transcription initiation from the Inr-containing promoter in WCE and reconstituted systems Inr in fission yeast (this work), and this requirement could be part of the recruitment of GTFs into the PIC and/or the recruitment of TICF.

The mechanism by which RNAPII is recruited into the TATA-less promoter has not yet been definitively elucidated. Our results support a model in which RNAPII is recruited at TATA-less promoters through interactions with TFIIB and TFIIF, as well as in the classic PIC formation on TATA-containing promoters; however, despite the PIC formation containing TBP-TFIIB-RNAPII/TFIIF-TFIIE-TFIIH, full-length transcripts have not been obtained. Perhaps that complex can polymerize the first 7–8 nucleotides and then a TICF is necessary to synthetize a longer RNA chain.

The requirement of PC4 in this in vitro system is not completely understood, although it acts as a positive cofactor through the Med and TFIIA in fission yeast [20]. PC4 can also be necessary to counterattack negative cofactors present in WCE and in the TCIF fraction. The gene encoding PC4 (sub1; SPAC16A10.02) is not essential in fission yeast cells, although its deletion causes loss of viability in G0 phase (Pombase). This suggests that it has an overlapping function with another gene(s), perhaps with a gene(s) encoding subunits of the Med complex, TFIIA, or another positive cofactor(s).

In mammals and *Drosophila*, there are different patterns of transcription initiation, which are referred to as focused (also known as narrow peak) and dispersed (also known as broad) [4,13,14]. Focused transcription occurs when transcription initiates from a single site or from a narrow cluster of initiation sites (five nucleotides or less). On the other hand, dispersed transcription occurs when there exists multiple transcription start sites, which can be spread over a region 50–100 nucleotides long [4]. Additionally, some promoters have a combined pattern of both focused and dispersed transcription initiation [4]. Often, the focused promoters contain core promoter elements, such as the TATA-box and the Inr element, and they are associated with regulated genes [4,14]. Dispersed promoters are associated with CpG-rich regions in mammals [13]. In fission yeast, the promoter architecture resembles that of mammalian promoters, and in this organism, it is expected that TATA-Inr and Inr-containing promoters would be mostly focused, in which transcription initiates from a single site or in a very narrow cluster. Indeed, the nmt1 promoter is a focused promoter, and transcription initiates from a main site (+1) in both TATA/Inr and Inr-containing promoters [16,18].

In conclusion, we found a promoter element in a subset of fission yeast gene promoters that is highly homologous to the initiator of the hDHFR promoter. This element is found in the nmt1 TATA-containing promoter, and it is able, in isolation, to direct complex formation and transcription initiation in a fission yeast WCE and in a reconstituted system. However, transcription initiation in the reconstituted system requires, in addition to the GTFs and RNAPII, PC4, Mediator, and TCIF, indicating that transcription initiation in vitro from initiator-containing promoters is more complex than transcription of TATA-containing promoters. Furthermore, a stable PIC on the initiator is formed by the GTFs and RNAPII, and the first step is the binding of TBP in association with either TFIIA or TFIIB. We did not find any TAF requirement, either for PIC formation or transcription initiation, which is different from the results of previous studies on metazoans. Our results suggest that TATA-containing and Inr-containing promoters use the same universal pathway for the assembly of the PIC.

## Figures and Tables

**Figure 1 genes-13-00256-f001:**
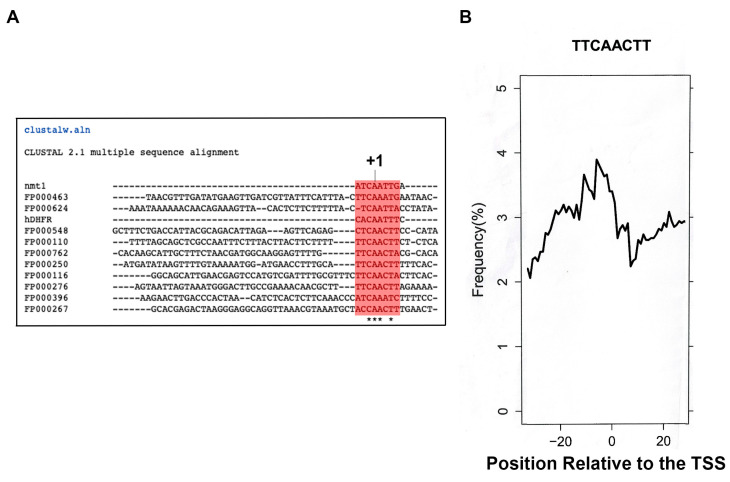
Fission yeast TATA-less promoters contain a highly conserved sequence around the TSS. (**A**) Several fission yeast TATA-less promoters (obtained from the EPD) were aligned with the nmt1 Inr and the human DHFR Inr. The conserved sequence (TTCA + 1ACTT), that represents the Inr, is highlighted in red. The alignment was performed with the ClustalW program (https://www.genome.jp/tools-bin/clustalw, accessed on 23 April 2021). (**B**) Distribution of the TTCAACTT sequence in *S. pombe* (obtained from the EPD). It can be observed that the exact match to this sequence is contained in approximately 4% of the fission yeast promoters described in *S. pombe* in the EPD (https://epd.epfl.ch, accessed on 12 June 2021).

**Figure 2 genes-13-00256-f002:**
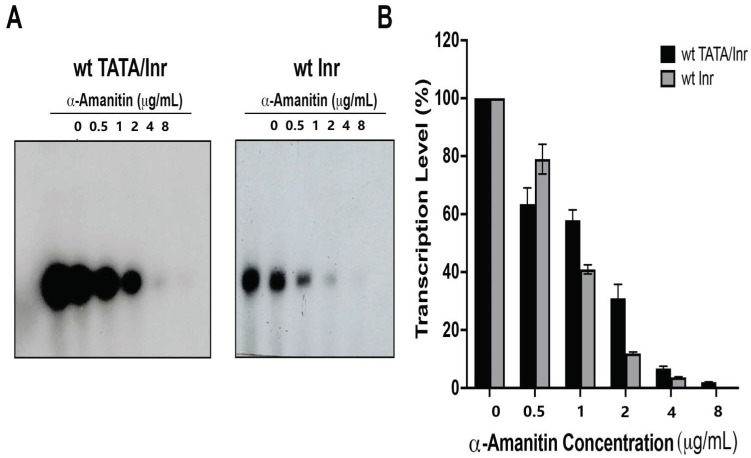
The nmt1 Inr is transcribed by RNAPII in WCE. (**A**) The nmt1 gene promoter containing a TATA box and an Inr is transcribed in WCE, and the transcription is inhibited by α-Amanitin. Concentrations of 0.5, 1.0, 2.0, 4.0, and 8.0 µg/mL were used to inhibit the transcription of the TATA box Inr-containing nmt1 promoter (wt TATA Inr). It can be observed that concentrations over 4.0 µg/mL of α-Amanitin completely inhibited the transcription of the wild-type TATA box Inr-containing promoter. On the other hand, concentrations of 2.0, 4.0, and 8.0 µg/mL of α-Amanitin completely inhibited transcription of the nmt1 Inr-containing promoter (wt Inr). Quantification of the experiments (*n* = 3) are shown in (**B**). The intensity of each transcription product was quantified using Image J software. Total pixels of each transcription product were measured and expressed as percentage related to the experiment without the addition of α-Amanitin, which was considered as 100% transcription.

**Figure 3 genes-13-00256-f003:**
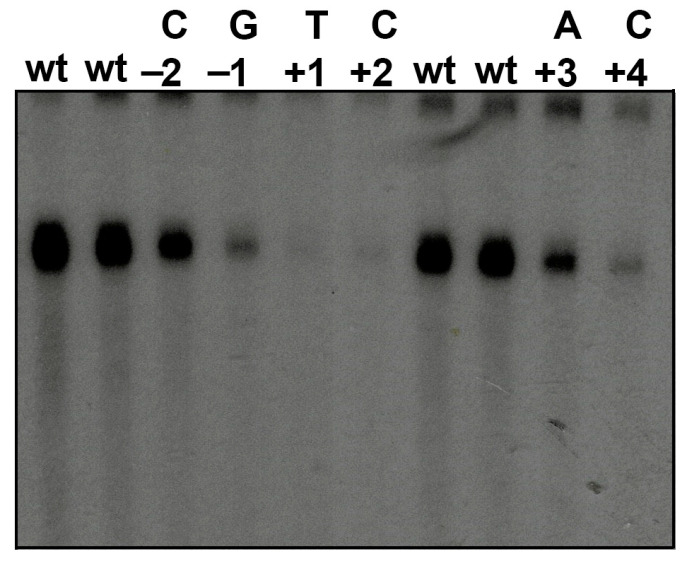
Mutations in conserved positions impairs transcription initiation. Those conserved positions were changed to bases, as indicated at the top of the figure. It can be observed that mutations on conserved positions at −1, +1, +2, and +4 impair transcription initiation of the nmt1 Inr-containing promoter (indicated as wt).

**Figure 4 genes-13-00256-f004:**
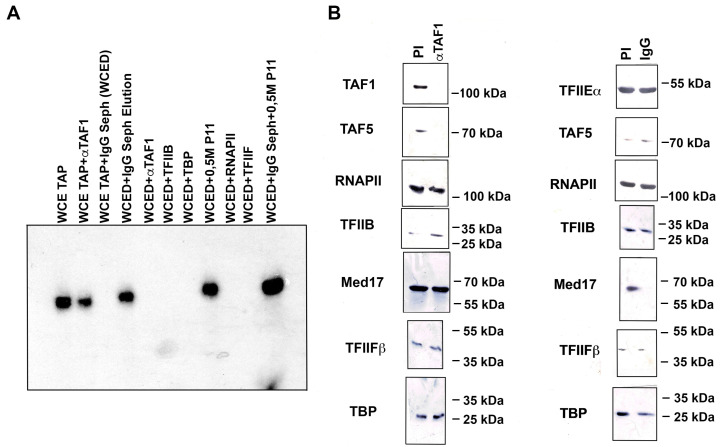
Mediator depletion inhibits nmt1 Inr transcription. (**A**) Mediator was depleted from a WCE from a TAP-tagged Med7 strain using IgG-Sepharose beads, and transcription was recovered by replacing different combinations of factors. The WCE from the same TAP-tagged strain was depleted of TAFs using anti-TAF1 antibodies crosslinked to Sepharose beads. It is observed that Mediator depletion completely abolishes transcription of the nmt1 Inr (WCED), and is recovered by replacing an eluate from the IgG-Sepharose beads (WCED + IgG Seph. Elution), with a crude fraction from a P11 chromatographic fraction of WCE (WCED + 0.5 M P11), or with the same fraction plus the elution of the IgG-Sepharose column (WCED + IgG Seph + 0.5 M P11). TFIIB, TBP, RNAPII, or TFIIF cannot complement for the activity eluted from the IgG-Sepharose beads. Importantly, TAF-depletion has no effect on nmt1 Inr transcription (WCE TAP + αTAF1) and an eluate from the anti-TAF1 beads cannot complement Mediator depletion (WCED + TAF1). (**B**) Western blot analysis with different antibodies was used to test the extension of the depletion of the WCED. It is observed that Mediator was completely depleted (Med17 panel) by IgG Sepharose beads, and TAFs are depleted by anti-TAF1 Sepharose beads (TAF5 panel). Depletion of Mediator does not affect the levels of TAFs, and depletion of TAFs does not affect Mediator levels. Levels of RNAPII and other transcription factors are not affected by either Mediator- or TAF-depletion.

**Figure 5 genes-13-00256-f005:**
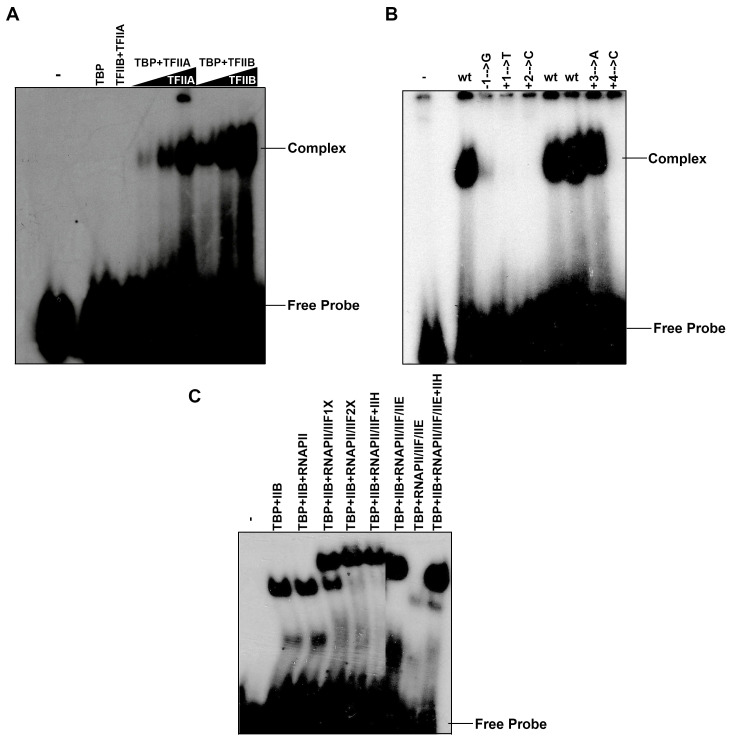
TBP binds to the nmt1 Inr in a complex, either with TFIIA or TFIIB. (**A**) TFIIA/TFIIB added together to a binding assay with a labeled probe containing the wild-type nmt1 Inr do not produce a detectable complex; however, when TFIIA or TFIIB are added together with TBP, complexes are obtained, indicating that TBP binds to the Inr in the presence of these factors. (**B**) Mutants of the nmt1 Inr, which are impaired in transcription, cannot form a TFIIB-TBP complex, indicating that these conserved positions are required for TFIIB-TBP complex formation. (**C**) Preinitiation complex formation on the nmt1 Inr follows a similar pathway as TATA box-containing promoters. It can be observed that TFIIB-TBP does not form a new complex with RNAPII in the absence of TFIIF; however, when RNAPII + TFIIF (RNAPIIIIF) are added, a new complex is formed that migrates slower than the TFIIB-TBP complex. Adding TFIIH to this complex does not produce any change in its mobility or amount; however, when TFIIE is added, the intensity of the complex augments, indicating that TFIIE was incorporated. A similar process occurs when TFIIH is added to the TFIIB-TBP-RNAPII-TFIIF-TFIIE complex. In the absence of TFIIB, RNAPII + TFIIF is not recruited into the complex, indicating that TFIIB makes a bridge between TBP and RNAPII + TFIIF. Figure 5C was constructed from two panels from the same Western blot.

**Figure 6 genes-13-00256-f006:**
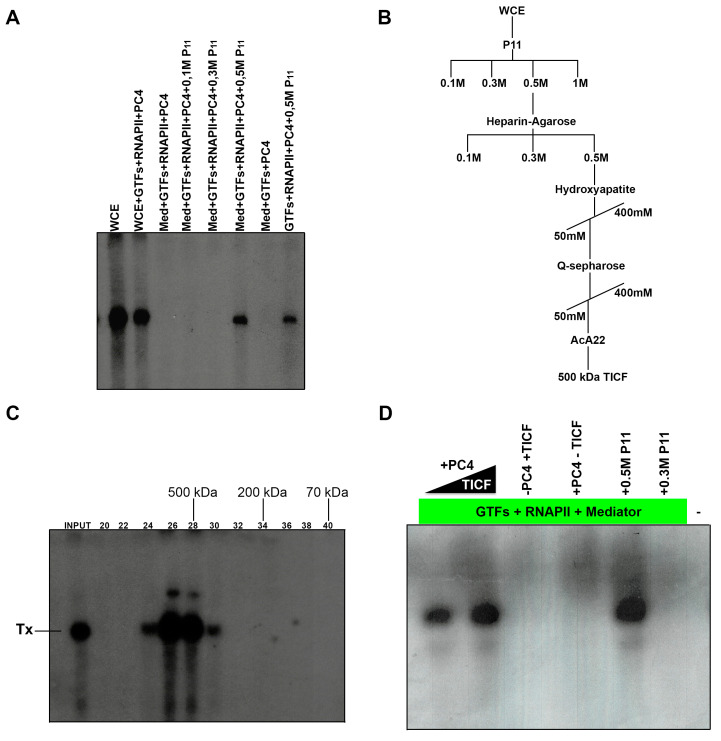
Nmt1 Inr transcription can be reconstituted with GTFs, RNAPII, Mediator, PC4, and a crude fraction from the 0.5 M P11 chromatographic column. A transcription assay for the nmt1 Inr was set up with GTFs, RNAPII, Mediator, and PC4; however, this purified system was not able to transcribe the nmt1 promoter. Addition of a crude chromatographic fraction from a P11 column (0.5 M P11) can support transcription in this assay. Addition of other chromatographic fractions from the P11 column (0.1 M and 0.3 M P11) do not support transcription from this promoter. (**A**) Omission of Mediator from the assay slightly diminishes transcription. (**B**) Chromatographic scheme of purification of TICF through several steps. The activity was followed by transcription assays set up with the nmt1 Inr template, GTFs, RNAPII, Mediator, and PC4. (**C**) Transcription assay and estimation of the molecular weight of TICF by gel filtration (AcA22 column). The molecular weight of TICF is near 500 kDa, as determined in the gel filtration column. (**D**) The activity of TICF is dependent on PC4. A transcription assay set up with GTFs, Mediator, and RNAPII can transcribe the nmt1 Inr when TICF and PC4 are added; however, when PC4 is absent from the assay (-PC4 + TICF), transcription no longer occurs. Consequently, when TCIF is omitted (+PC4-TCIF), transcription does not occur. The crude chromatographic fraction of 0.5 M P11 can replace TICF and PC4 (+0.5 M P11), however, the 0.3 M P11 fraction cannot replace the activities (+0.5 M P11). Tx indicates the transcription product.

**Figure 7 genes-13-00256-f007:**
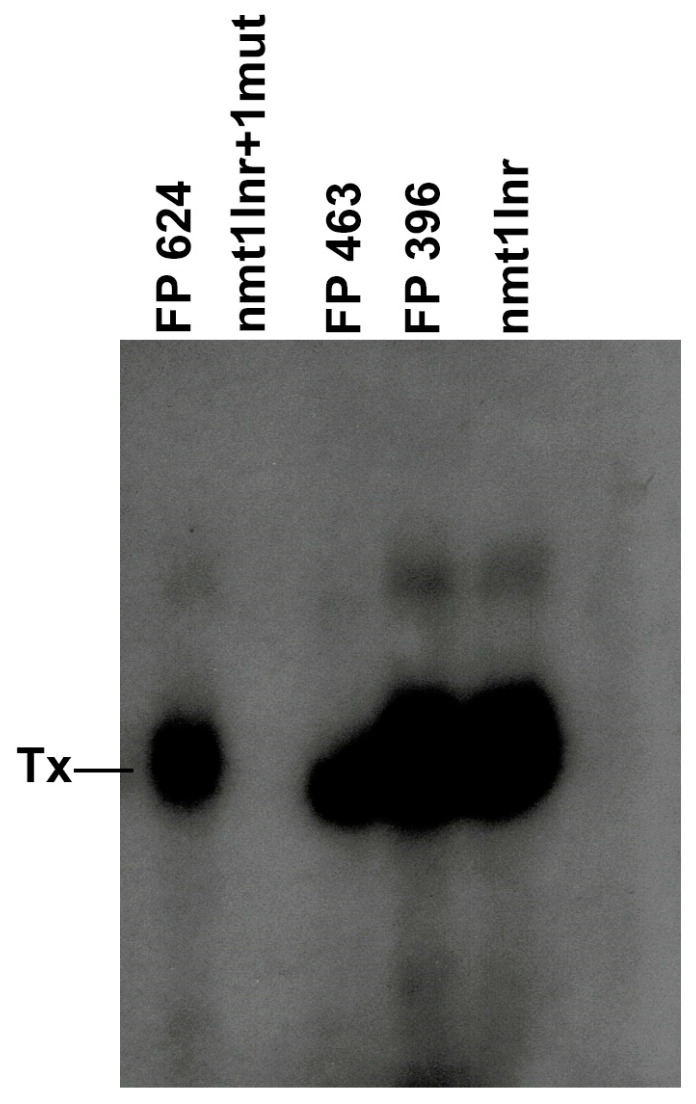
Inr of the hDHFR/nmt1 family can be transcribed in WCE. Three promoters, namely FP000624 (FP 624), FP000463 (FP 463), and FP000396) (FP 396) were cloned (from −20 to +10), fused to the G-less cassette, transcribed in WCE, and compared with the nmt1 Inr +1 mutant (nmt1Inr + 1mut) and the wild-type nmt1 Inr (nmt1Inr). It can be observed that the three promoters can support transcription initiation in crude extracts. Tx indicates the transcription product.

## Data Availability

This study did no report any data.

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
