# Peer review of "Initiator-Directed Transcription: Fission Yeast Nmtl Initiator Directs Preinitiation Complex Formation and Transcriptional Initiation"

_genes, 2022, doi:10.3390/genes13020256_

Round 1

Reviewer 1 Report

The Initiator element is a core promoter element, which is observed in TATA-less promoters. The mechanism of the Initiator-dependent transcription initiation is still not well understood. In this manuscript, the authors identified a conserved sequence in fission yeast TATA-less promoters as a potential Initiator with similarity to the Initiator of the hDHFR. This conserved sequence is also detected in the nmt1 promoter, which contains a classical TATA-box. The authors used the nmt1 promoter to investigate the importance of this potential Initiator sequence and examined what components were required for the nmt1 Initiator-dependent transcription.

 The research topics are important, and the presented data are interesting. However, there are some problems with the experimental design and the presentation of the results.

  1. This reviewer does not understand why the authors did not examine the effect of α-Amanitin at the concentrations of 0.5 and 1.0 ug/ml for the wt Inr as for the wt TATA/Inr in Figure 2. It is better to show that there is a concentration dependence of α-Amanitin in inhibition of transcription of the nmt1 Inr-containing promoter.
  2. In all figures, especially Figure 4, frames around the blots are needed. The boundaries of each blot are unclear. Further, the authors should present all blots with at least one molecular weight marker.
  3. In Figure 5C, TBP bound to the nmt1 Inr without IIB when mixed with RNAPII/IIF/IIE+IIH, which seems to be inconsistent with the authors’ claim that TFIIA or TFIIB is necessary for the binding of TBP to the nmt1 Inr.
  4. Figure 5C is created by combining two blots. Consolidation of data must be made apparent and should be indicated in the figure legends.
  5. There is no description about what “TSP” at line 168 and “Tx” in figures 6C and 7 stand for.
  6. Lines 380-384 should be omitted.

Author Response

Reviewer 1

First, we would like to thank for the careful and critical revision of our manuscript. Those comments will certainly improve the contents of the submitted manuscript.  Below you can find the answers to each comment, and we hope that now the manuscript would be ready for publication.

1. This reviewer does not understand why the authors did not examine the effect of α-Amanitin at the concentrations of 0.5 and 1.0 ug/ml for the wt Inr as for the wt TATA/Inr in Figure 2. It is better to show that there is a concentration dependence of α-Amanitin in inhibition of transcription of the nmt1 Inr-containing promoter.

RESPONSE: We have changed the figure for another duplicate experiment, in which we assayed the same concentrations of a-amanitin on both templates. Likewise, from preliminary experiments we learnt that with 4 and 8 ug/ml of a-amanitin inhibits more than 95 % of the transcription from the wt Inr template. In addition, we have added a quantification of the experiment to clearly demonstrate a dose-dependence of the RNA pol II inhibitor.

2. In all figures, especially Figure 4, frames around the blots are needed. The boundaries of each blot are unclear. Further, the authors should present all blots with at least one molecular weight marker.

RESPONSE: We have added frames to all the blots and the boundaries of blot panels in the Figure 4 were incorporated. In addition, molecular weight markers were added to the blot panels of Figure 4B.

3. In Figure 5C, TBP bound to the nmt1 Inr without IIB when mixed with RNAPII/IIF/IIE+IIH, which seems to be inconsistent with the authors’ claim that TFIIA or TFIIB is necessary for the binding of TBP to the nmt1 Inr.

RESPONSE: Thank you very much for that observation. We are extremely sorry for the mistake, however that is an editing error, since IIB was not included in the label, but indeed TFIIB was added to the reaction to form the complete preinitiation complex with the GTFs and RNA pol II. The correct label is TBP+IIB-RNAPII/IIF+IIE´+IIH. We have enclosed the raw original figure, where it clearly can be seen that TFIIB was included in that particular reaction. It must be noted that inclusion of TFIIH to the TBP+TFIIB+RNAPII/IIF+IIE complex does not alter the mobility of the complex, although it augments the amount of complex. To get a very slight shift in the presence of TFIIH, ATP must be included in the reaction mix to phosphorylate the CTD of RNAPII by TFIIH.

4. Figure 5C is created by combining two blots. Consolidation of data must be made apparent and should be indicated in the figure legends.

RESPONSE: Fig 5C was not composed from two different films. It was created with two panels from the very same film, and there are experiments in which three lanes containing the complex were assayed with high amounts of TFIIB. However, those resulting complexes were very intense and did not separate enough. For that reason, those three lanes were cut down and taken off to avoid any confusion. Then, the two additional lanes were combined to the figure. As evidence of that we enclosed the raw original film for the reviewer, and we indicated that in the figure legend.

5. There is no description about what “TSP” at line 168 and “Tx” in figures 6C and 7 stand for.

RESPONSE: We have changed and corrected “TSP” by “TSS”, which has been defined in the text as “Transcription Start Site”. Tx was defined as the transcription product.

6. Lines 380-384 should be omitted.

RESPONSE: We have eliminated those lines in the new version.

Reviewer 2 Report

In the manuscript "Initiator directed transcription: Fission yeast nmtl initiator directs preinitiation complex formation and transcriptional initiation," Rojas et al. investigate requirements for transcription initiation in a truncated version of Schizosaccharomyces pombe nmtl promoter. The manuscript is well written and logical, with conclusions presented in it being supported by the results. However, there are several points the authors should address:

1) Figure 1A: The authors specify that all eight bp of the conserved sequence are highlighted in red, but they highlighted only four bp, those absolutely conserved ones. The authors should resolve the discrepancy.

2) Sections 3.3. and 3.7. describe results obtained with mutated constructs, but Material and methods do not include any details on molecular techniques and primers used in the construction. Please, include all relevant details in the Materials and methods.

3) The authors should discuss how TATA-box promoters compare to Initiator-only promoters in terms of narrow and broad TSS clusters (e.g. see https://doi.org/10.1038/ng1789)

4) One final additional paragraph serving as a conclusion would be welcome.

5) Keywords are missing.

6) Lines 380-384, 485-509 are superfluous and should be removed.

7) There are several misspelt words:

line 12: Espana > Spain
line 42: gen > gene
line 43: for transcription the > for transcription of the
line 92: nmt1transcription > nmt1-transcription
line 205: probably due to is rather > sentence is unclear, please reformulate
line 235: observed in vivo conditions > observed in in vivo conditions
line 326-328: the sentence is unclear, please reformulate
line 422: stablish > establish
line 455: us > as

Author Response

First, we would like to thank  for the careful and critical revision of our manuscript. Those comments will certainly improve the contents of the submitted manuscript.  Below you can find the answers to each comment, and we hope that now the manuscript would be ready for publication.

1) Figure 1A: The authors specify that all eight bp of the conserved sequence are highlighted in red, but they highlighted only four bp, those absolutely conserved ones. The authors should resolve the discrepancy.

RESPONSE: Thank you very much for that observation. In the new manuscript version, Figure 1A has been corrected and the highlighted sequence was expanded to eight bp.

2) Sections 3.3. and 3.7. describe results obtained with mutated constructs, but Material and methods do not include any details on molecular techniques and primers used in the construction. Please, include all relevant details in the Materials and methods.

RESPONSE: Thank you very much for that suggestion. In Materials and Methods of the new version we have incorporated the detail of all techniques used to generate mutated version of the promoters which were evaluated in the work.

3) The authors should discuss how TATA-box promoters compare to Initiator-only promoters in terms of narrow and broad TSS clusters (e.g. see https://doi.org/10.1038/ng1789)

RESPONSE: We have added at the end of the discussion a new paragraph where the authors compare the differences between narrow and broad TSS clusters in the different gene promoters. We speculate that TATA-Inr and Inr-only promoters in fission yeast should have features comparable to focused promoters and transcription should initiate at a narrow cluster in most of these promoters.

4) One final additional paragraph serving as a conclusion would be welcome.

RESPONSE: Thank you very much for the suggestion. We have added one additional paragraph at the end of the Discussion section with a main conclusion of our work,

5) Keywords are missing.

RESPONSE: Keywords were added to the new version of the manuscript.

6) Lines 380-384, 485-509 are superfluous and should be removed.

RESPONSE: The text associated to those lines was removed in the new version.

7) There are several misspelt words:

line 12: Espana > Spain
line 42: gen > gene
line 43: for transcription the > for transcription of the
line 92: nmt1transcription > nmt1-transcription
line 205: probably due to is rather > sentence is unclear, please reformulate
line 235: observed in vivo conditions > observed in in vivo conditions
line 326-328: the sentence is unclear, please reformulate
line 422: stablish > establish
line 455: us > as

RESPONSE: Thank you very much for those observations. We have corrected all the misspelt words and we have reformulated lines 205 and 326-328. Additionally, the text was corrected to avoid grammatical mistakes.